# Clustering Inflammatory Markers with Sociodemographic and Clinical Characteristics of Patients with Diabetes Type 2 Can Support Family Physicians’ Clinical Reasoning by Reducing Patients’ Complexity

**DOI:** 10.3390/healthcare9121687

**Published:** 2021-12-06

**Authors:** Zvonimir Bosnic, Pinar Yildirim, František Babič, Ines Šahinović, Thomas Wittlinger, Ivo Martinović, Ljiljana Trtica Majnaric

**Affiliations:** 1Department of Internal Medicine, Family Medicine and the History of Medicine, Faculty of Medicine, Josip Juraj Strossmayer University of Osijek, 31000 Osijek, Croatia; zbosnic191@gmail.com; 2Department of Computer Engineering, Faculty of Engineering and Natural Science, Istanbul Okan University, 34959 Istanbul, Turkey; pinar.yildirim@okan.edu.tr; 3Department of Cybernetics and Artificial Intelligence, Faculty of Electrical Engineering and Informatics, Technical University of Košice, 04201 Košice, Slovakia; frantisek.babic@tuke.sk; 4Department of Clinical Laboratory Diagnostics, Osijek University Hospital Centre, 31000 Osijek, Croatia; ines.sahinovic@kbco.hr; 5Department of Cardiology, Asklepios Hospital, 38642 Goslar, Germany; dr.wittlinger@gmx.de; 6Department of Cardiothoracic Surgery, University Hospital Marburg, 35033 Marburg, Germany; ivo.martinovic@med.uni-marburg.de; 7Faculty of Dental Medicine and Health Osijek, J.J. Strossmayer University of Osijek, 31000 Osijek, Croatia

**Keywords:** diabetes type 2, chronic inflammation, complex chronic diseases, primary care patients, phenotyping, data mining, clustering techniques

## Abstract

Diabetes mellitus type 2 (DM2) is a complex disease associated with chronic inflammation, end-organ damage, and multiple comorbidities. Initiatives are emerging for a more personalized approach in managing DM2 patients. We hypothesized that by clustering inflammatory markers with variables indicating the sociodemographic and clinical contexts of patients with DM2, we could gain insights into the hidden phenotypes and the underlying pathophysiological backgrounds thereof. We applied the k-means algorithm and a total of 30 variables in a group of 174 primary care (PC) patients with DM2 aged 50 years and above and of both genders. We included some emerging markers of inflammation, specifically, neutrophil-to-lymphocyte ratio (NLR) and the cytokines IL-17A and IL-37. Multiple regression models were used to assess associations of inflammatory markers with other variables. Overall, we observed that the cytokines were more variable than the marker NLR. The set of inflammatory markers was needed to indicate the capacity of patients in the clusters for inflammatory cell recruitment from the circulation to the tissues, and subsequently for the progression of end-organ damage and vascular complications. The hypothalamus–pituitary–thyroid hormonal axis, in addition to the cytokine IL-37, may have a suppressive, inflammation-regulatory role. These results can help PC physicians with their clinical reasoning by reducing the complexity of diabetic patients.

## 1. Background

Chronic inflammation is considered the main mechanism in aging and the development of chronic diseases, including diabetes mellitus type 2 (DM2), cardiovascular disease (CVD), neurodegenerative disease, and some malignant and musculoskeletal diseases [1]. It has been recognized that the impact of environmental factors is more important than genetics in directing the course of aging towards “unsuccessful aging”, that is, aging burdened with chronic diseases and functional impairments [2,3]. There are numerous sources of inflammation in older individuals, with cell senescence and chronic activation of the innate immune receptors being the main ones [3]. Changes in the body’s shape and structure that occur with aging, including muscle loss and an increase in visceral fat, significantly contribute to inflammation and the development of insulin resistance, which taken together increase the risk for metabolic and vascular disorders [4]. Obesity accelerates these age-related processes [4]. It has become increasingly clear that age-related inflammation is a whole-body response, which involves a variety of brain–immune and gut–brain feedback loops [2]. Many compensatory and anti-inflammatory mechanisms, still insufficiently known, may counteract the deleterious effects of chronic inflammation on survival and functional decline of older individuals [1,5].

DM2 is a complex chronic disease that involves metabolic, inflammatory, and vascular disorders [6]. Because of its global spread and serious consequences, DM2 represents a major public health concern [7]. Notably, the risk of CVD as a leading cause of mortality is several times higher in these patients than in the general population [8,9]. Initiatives are emerging for a more personalized approach in managing patients with DM2 [10]. In addition, efforts have been undertaken to develop new treatment strategies. Instead of focusing on antihyperglycemic treatment effects, as was the case in old-fashioned medications, the focus now is on CV benefits that go beyond glucose-lowering effects [11].

For risk stratification and customized treatment of diabetic patients, it is important to know that CVD, DM2, and comorbidities such as hypertension, metabolic syndrome (a cluster of disorders associated with visceral obesity), and chronic kidney disease (CKD) have a common pathophysiological background [4,12,13]. Moreover, because of the current understanding of inflammation as the common driver of age-related chronic diseases, other comorbidities that cluster together with DM2 and CVD should be taken into consideration as well [14]. This is also the case for mental disorders, anxiety, and depression, which are known to often accompany DM2 and CVD, worsening the course of these conditions [15]. Recent evidence indicates that frailty, if associated with chronic health conditions, can modify CV risk factors and influence the outcomes [16,17,18]. The state of frailty is manifested by a reduction in muscle mass and strength, slow walking, and low activity and is considered reflective of exhausted homeostatic reserves in multiple organs and systems [19].

Inflammatory markers such as the cytokine IL-6 and the acute-phase reactant C-reactive protein (CRP) have long been used to stratify risk in patients with CV risk factors [20]. Recently, however, a more prevalent role has been given to the neutrophil-to-lymphocyte ratio (NLR) [21]. NLR represents a low-cost and widely available laboratory parameter and is suitable for community-based research. In addition, it reflects tissue-related inflammatory processes associated with the progression of atherosclerosis and end-organ damage [22,23,24,25]. In both cases, increased recruitment of inflammatory and immune cells from the circulation to the tissue via dysfunctional vascular endothelial cells constitutes the main underlying mechanism [26,27]. Distinctly from atherosclerosis, in which large arteries are the target of an inflammatory response, microcirculation plays a major role in tissue-related cell infiltration, which occurs during the progression of end-organ damage. However, before NLR can be used routinely, it needs to be validated in different populations, as many sociodemographic and lifestyle factors can influence variations in its values [28]. The relevant reference values need to be identified not only for the general population, but also to distinguish between particular cohorts or to indicate specific adverse outcomes [29].

Some new markers of inflammation have been explored for their potential use for risk stratification in everyday clinical practice. A very promising one is the proinflammatory cytokine IL-17A [30]. For its effect on mobilization of inflammatory cells from the circulation to the tissues, particularly relating to neutrophils and Th17 lymphocytes, the main effector cells in tissue-related chronic inflammation, this cytokine plays a central role in the process of end-organ damage and is complementary to NLR [31]. The role of this cytokine in promoting diabetic CKD has been clearly confirmed in experimental studies [32]. Another promising inflammatory marker, yet unexplored in real-life settings, is cytokine IL-37, which has been recognized for its broad anti-inflammatory and glucose homeostasis modulating effects [33,34]. This cytokine was found to be directly involved in reducing high-glucose-induced inflammation and oxidative stress as well as organ damage caused by these mechanisms [35]. Both cytokines proposed in this study have a chance to be used as new biological treatments. If this happens, it will constitute a major turn in medication therapy that can be used for preventing CVD in patients with DM2 [36]. Moreover, the possibility to detect gene polymorphisms for these cytokines may prompt personalized treatment strategies [37].

## 2. Aim

Currently, there is no consensus as to which markers of inflammation best represent chronic inflammation or can distinguish among the various phases of inflammatory responses [38]. The fact that DM2 is mostly an age-related disease accentuates heterogeneity of DM2-related phenotypes by increasing the potential in DM2 patients for the development of multiple comorbidities, malnutrition, sarcopenia, and frailty [39,40]. In addition to the well-known fact that the risk of vascular complications and mortality increases with the duration of DM2 and is dependent on how glycemia is regulated, new evidence has also emphasized, in this regard, the importance of patient age and the age of DM2 onset [9,41]. Guided by these facts, we hypothesized that by clustering inflammatory markers with variables indicating sociodemographic and clinical characteristics of patients with DM2, we could gain insights into the hidden phenotypic subtypes that exist in primary care (PC) diabetic patients in a real-life setting and in inflammation-related pathophysiology networks that stay beyond these phenotypes. The proposed research approach was expected to help PC physicians with clinical reasoning by reducing the diagnostic uncertainty that PC physicians deal with on a daily basis when facing complexity of patients with DM2 and other chronic age-related diseases [42].

## 3. Methods

### 3.1. Study Population

The study was performed in a PC setting in the area of Slavonski Brod (59,000 residents), a town in the southeastern part of Croatia. In Croatia, family physicians have a gatekeeping role, which enables good access to the general population. PC services are supported by an IT system [43]. Chronic disease surveillance and preventive check-up platforms were established as a part of electronic health records (eHRs) to improve screening and management strategies for patients with some important chronic conditions, including DM2. Despite the established eHRs, patient data are still extracted manually, however, as eHRs in individual family medicine (FM) practices are not integrated into the common research platform. Our study patients were recruited from four FM practices in which family physicians allowed access to medical records. The study was approved by the Expert and Ethics Council of the Health Center of Slavonski Brod (ID:1433-1/020).

Patients aged 50 years and above of both genders and diagnosed with DM2 were recruited into the study. At these ages, obesity, metabolic disorders, and DM2 also start to emerge at higher rates [44]. In the FM practices included in the study, 1600–1800 patients were registered in the lists, giving a source population of 6500–7000. About a quarter of patients (400–450 patients per one FM practice) were from the target age groups. The number of patients per FM practice who were diagnosed with DM2 amounted to 80–100, with the prevalent number being from the target age groups.

As selection criteria, patients had to be able to independently visit their family physicians and give written informed consent. Patients with acute conditions or in rapidly declining health and those with malignant diseases who had received chemo- or radiotherapy or biological treatments were excluded. Also excluded were individuals with an amputated leg, those on continuous renal replacement therapy or with a transplanted kidney, and those with dementia or significant cognitive dysfunction.

The study was conducted over a period of four months, from September to December 2020. Of the initially selected 228 eligible patients, 190 patients finished the study protocol. During the study period, the first COVID-19 pandemic wave was underway, and mass vaccination had not yet been introduced, which hindered study execution. We were also limited by the package size of the cytokine research reagents, so the final number of patients for whom a full set of data was available, as planned for research, was 174.

### 3.2. Data Collection

We used a total of 30 variables, of which 4 were markers of inflammation and all of the others were patient sociodemographic and health status characteristics (Table 1). 

Two markers of inflammation are routinely used in the PC setting, CRP and hemoglobin (Hb), a marker of anemia of inflammation and chronic diseases [45]. We also included some emerging markers of inflammation in the study; we propose that NLR and the cytokine IL-17A better reflect tissue-related chronic inflammation and that cytokine IL-37 indicates the anti-inflammatory response.

Blood samples were taken from patients for laboratory testing. All tests were performed in the local hospital’s central laboratory, except the analysis of cytokines, for which blood specimens were sent to the laboratory for molecular immunology of the University Hospital Clinical Center in Osijek, the administrative and university center of this region. Serum lipids, triglycerides, and HDL cholesterol were included in the dataset, as they indicate metabolic syndrome, a metabolic disorder associated with obesity, hypertension, and DM2 [9]. We did not include LDL cholesterol, as many patients were treated with hypolipidemic drugs, and serum concentrations of this lipid fraction were significantly modified by this treatment. We needed serum creatinine tests to estimate the glomerular filtration rate (eGFR), a measure of renal function decline [46]. According to the international classification, there are 4 levels of renal function decline, where levels 1 and 2 are associated with preserved function and levels 3 and 4 with lowered function. This measure was calculated using an online calculator [47]. Variations in TSH values can reveal age-related dysregulation of the hypothalamus–pituitary–thyroid axis, which is known to be implicated in metabolic and CV disorders in older individuals, in particular those with multiple comorbidities [48]. TSH levels in the range of 4–10 mU/L indicate latent hypothyroidism, a common age-related disorder, but still not in need of treatment. The NLR was calculated from the complete blood count with differential [49].

From eHRs, we used data indicating patient sociodemographic characteristics, comorbidities, and treatment options for DM2. We took into consideration only comorbidities known to be closely associated with DM2, such as hypertension and CVD, and which frequently affect diabetic patients or significantly contribute to inflammation, such as the common musculoskeletal disorders and anxiety/depression [50]. Of currently available treatment options for DM2, we included insulin therapy for its known anti-inflammatory effect and new antidiabetic medications for their favorable CV effects [51,52].

Anthropometric measurements of body mass index (BMI), indicating general obesity, and midarm circumference (mac), indicating muscle loss, were performed in patients at their visits [53]. Patients were also assessed for physical frailty using Fried’s phenotype model [19]. This model takes into account five predefined and easily measured criteria, weight loss, feeling of exhaustion, low physical activity, slow walking, and weakness (low grip strength, measured by hand dynamometer). Patients were classified by the number of positive criteria as being frail (≥3), prefrail (1–2), or robust (none).

Although medical science is disease based, there is no adequate way to integrate the coexisting diseases in complex patients [54]. For creating clusters, we used the item “the number of comorbidities” (<3 vs. ≥3) rather than CIRS (the Cumulative Illness Rating Scale), today the most used weighted scoring system for measuring comorbidity-related complexity [55]. This instrument comprehensively evaluates impairments in different organs and systems, depending on their levels of severity, but without using specific diagnoses of diseases. In this study, there was no need to comprehensively assess all possible comorbidities that a patient had. Instead of that, the aim was to explore the most common comorbidities of patients with DM2 that are, at the same time, associated with increased inflammation. In doing so, we relied on both, evidence indicating CVD as the most prevalent comorbidities in patients diagnosed with DM2 and information indicating the most common non-CV comorbidities that can be found in patients with DM2 in our surrounding area [50].

In addition, CIRS was validated for some specifically defined outcomes, such as hospital-related mortality and the length of hospital stay [55]. This study had a cross-sectional design and the aim to find out the reasonable phenotypic patterns of patients with DM2 that may be hidden within the dataset.

Moreover, although CIRS takes into account a contribution of each organ impairment to the comorbidity burden, it cannot discern among particular conditions, as it uses aggregate measures of the employed items [55]. A simple count of comorbidities, on the contrary, does not account on the severity of particular disease. This was, in our study, to some extent compensated by the involvement of some other clinical variables, such as laboratory tests, anthropometric measures, and frailty status, which helped us stratify patients into clusters according to levels of severity.

Furthermore, CIRS is based on the assumption that diseases and impairments do not associate with each other, while it is known that disease clustering (nonrandom associations) and disease interactions are features inherent to multimorbidity and complexity [56]. Imbalanced distribution of particular comorbid conditions among the clusters, in this study, revealed the nonrandom associations of these conditions with each other and with other features that were used to create the clusters, of which some contained prognostic elements (for, e.g., DM2 duration and the time of onset) [56,57]. In this way, the applied research approach incorporated many aspects of the concept of complex patients, such as different types of features interacting with each other, staging of severity, treatment difficulties, nonlinear associations (a small difference in one variable leading to a large difference in another variable), and feature patterns being inherently associated with phenotypes, which to some extent have a prognostic capacity [54,57].

### 3.3. Statistical Analysis

Numerical variables were presented as a range (min–max) and the arithmetic mean. Categorical variables were presented as frequencies of predefined categories (Table 1). The k-means algorithm, a machine-learning method, was applied to the dataset to search for hidden clusters [58]. In addition, we used multivariate linear regression (MLR) models to assess the statistically significant correlations between the variables NLR, Il-17A, and Il-37, indicating chronic inflammation (used as dependent variables), and all other variables, indicating patient contexts (used as independent variables). Statistical analyses were performed using WEKA 3.9.5 software, an open-source software consisting of machine-learning algorithms for data-mining tasks. It contains some tools for data preprocessing, classification, regression, clustering, and association-rules mining [59].

### 3.4. Cluster Analysis

Cluster analysis constitutes one of the main methods for data-mining studies [60]. The process of separating many objects into classes is known as clustering. A cluster is a collection of data that are similar or close to each other and far from data in other clusters. Cluster analysis is widely used in a variety of applications, including pattern recognition, data analysis, image processing, and biomedical research.

There are different clustering methods [61]. In this study, we used the k-means algorithm, a simple and effective method for implementation [62]. It is one of the so-called unsupervised learning methods, which means that knowledge of the outcome is not specified in advance. Such methods are used when undefined patterns or clusters need to be found in a dataset or to reduce the number of variables.

The k-means algorithm requires that the number of clusters (k) in the data is prespecified [63]. Finding the appropriate number of clusters for a given dataset is often difficult, and it takes many trials to decide which clustering option is “correct”. There is a relationship between performing the algorithm and choosing the k-value. However, studies on k-means clustering do not give much information on how to choose the k-value [64]. Implementation of the k-means algorithm in some software packages and tools requires the number of clusters to be specified by the user. To achieve high accuracy for the model, the algorithm must be run with many k-values and the results observed.

### 3.5. Data Analysis Procedure

We applied the k-means algorithm for different numbers of clusters (3 ≤ k ≥ 6) and observed the results of the sum of squared errors and run time for both clustering studies. The total squared error (SSE) shows how similar data in a set are to each other [63]. We compared the results of different numbers of clusters generated and found that the error decreased as the number of clusters increased. For example, when the value of k was 3, the SSE was 2712, and when the value of k increased to 6, the new value of the SSE was 2592 (Table 2). 

We obtained two results based on 3 and 6 clusters (Table 3 and Table 4). 

The MLR models with the markers NLR, Il-17A, and Il-37, used as the target outcome variables, are presented in Table 5.

## 4. Results

In the group of 174 examined PC diabetic patients, males were more prevalent than females (Table 1). According to the selection criteria, patients were aged 50 years or above, and some of them were of very old age (over 85 years). Patients varied highly in number of years of having DM2. Most of them had hypertension, but the duration of hypertension varied greatly. Other major characteristics were high number of comorbidities (≥3 in most) and high variations in anthropometric measures and laboratory tests, also including markers of inflammation. Serum concentrations of cytokine IL-37 were in the range between 0.14 and 258.8, except for one patient, in whom a serum concentration of 1788.4 pg/mL was measured.

With respect to comorbidities, CVD was most prevalent; almost half of the patients had a diagnosis of CHD, and one-third of them were diagnosed with CAD. Musculoskeletal diseases were also common, and every second patient had markedly developed osteoarthritis. Mood disorders, diagnosed as anxiety/depression, were also highly prevalent, presenting in more than a third of patients. Patients were distributed across all levels of renal function. A high proportion of patients lacked physical robustness, showing signs of prefrailty or frailty.

Relatively small proportions of the examined patients had used some of the new-generation antidiabetic medications, and less than a quarter of patients were under therapy with insulin.

It can be seen from Table 2 that the accuracy of the clustering procedure was higher in models with 5 or 6 clusters than in those with lower numbers of clusters. The user (family physician) made a decision about which one of two clusters with equal performance was more reliable from the clinical point of view. 

At least three clinically interpretable phenotypes could be recognized in the group of examined diabetic patients (Table 3). Two phenotypes (clusters Cl 1/3 and Cl 2/3) referred to obese individuals (BMI > 30) aged about 60 years; however, the first phenotype (cluster Cl 1/3) was dominated by men, whereas the second (cluster Cl 2/3) was dominated by women. Patients in these two clusters also differed from each other in the time when hypertension and DM2 began to emerge. In the dominantly male group, hypertension started in middle age, and DM2 developed recently (at the age of about 60). The dominantly female group had recently acquired both hypertension and DM2.

In contrast to these two clusters, cluster Cl 3/3 was dominated by men of older age (about 70 years) who were not obese and who had had both hypertension and DM2 for a long period of time. Only patients in this cluster were frail, predominantly had atherosclerotic CVD (CAD and PAD), and had significantly decreased renal function (at stage 3).

It can be seen in Table 3 that the weighted values of the laboratory tests, triglycerides, HDL cholesterol, TSH, hemoglobin, and CRP varied across the clusters. Neither of the considered therapy options was prevalently used in any of the three clusters.

A condensed presentation of clinical profiles of individuals selected from particular clusters in the three-cluster model is provided in Figure 1.

In the clusters identified by the three-cluster model, patterns of the inflammatory markers NLR, IL-17A, and IL-37 also differed from each other (Figure 2). For all of these markers, serum concentrations were higher in patients in cluster Cl 2/3 than in patients in cluster Cl 1/3. When clusters Cl 2/3 and Cl 3/3 were compared with each other, the opposite directions for cytokines IL-17A and IL-37 could be observed. While the values for cytokine IL-17A were lower, the values of cytokine IL-37 were higher in cluster Cl 3/3 than in cluster Cl 2/3. Cluster Cl 3/3 was curious in that serum concentrations of cytokine IL-37 were markedly increased.

It was evident from the six-cluster model that a variety of phenotypes may exist among patients with DM2 (Table 4). When comparing the patterns of the sociodemographic and clinical characteristics of patients in the clusters, indicated by age, gender, CV comorbidities, levels of renal function decline, and hypertension and DM2 duration and time of onset, some new knowledge is provided. For example, incident DM2 occurs at the age of around 60, but came in several subtypes (clusters Cl 2/6, Cl 5/6, and Cl 6/6), depending on the time when hypertension developed and on whether men or women were predominant in the cluster. Other phenotypes referred to patients with late-onset DM2 (in their 70s), such as those in cluster Cl 3/6, or to patients with long-lasting DM2 (clusters Cl 1/6 and Cl 4/6), which profiles nevertheless differed from each other based on differences in the prevailing age, gender, and time of DM2 onset.

By further comparing the six clusters, we could see that fairly low renal function (stage 3) could be expected in diabetic patients of advanced age (in their 70s and 80s) and in those with long-term hypertension duration (clusters Cl 3/6 and 4/6). Only these two oldest groups of diabetic patients were also featured with multiple CV comorbidities. Although similar in many features, these two patient groups yet differed from each other in an important aspect of health, the frailty status.

As in the three-cluster model, in the six-cluster model, there was a lack of variation in variables indicating therapeutic options.

A condensed presentation of the sociodemographic and clinical profiles of individuals selected from particular clusters in the six-cluster model is provided in Figure 3. 

A graphical presentation of the inflammatory markers NLR, IL-17A, and IL-37 across the clusters is provided in Figure 4.

When comparing the clinical and sociodemographic contexts and patterns of the inflammatory markers NLR, IL-17A, and IL-37 in the clusters identified by the three- and six-cluster models, we could draw some parallels and find out some new cluster subtypes that were hidden in the three-cluster model (Figure 5).

It can be seen from Figure 5 that a separation of the “parent” clusters into new cluster subtypes was not consistent. The clusters that stayed in their “parent” clusters when the k-value increased were those dominated by the male gender (the pairs Cl 1/3–Cl 2/6 and Cl 3/3–Cl 4/6). The cluster Cl 2/3 in the three-cluster model, which refers to women in postmenopause and with recent-onset DM2, spread out in the six-cluster model into two new subtypes (Cl 5/6 and Cl 6/6), of which the first represented women in early postmenopause and the second represented women in late postmenopause. Accordingly, these two new cluster subtypes differed from each other in hypertension duration. There were also two new cluster entities that were revealed when the k-value increased from three to six. One of them (Cl 1/6) indicated women in postmenopause but with long-lasting DM2, and the other (Cl 3/6) indicated women of older age (about 70 years) with recent-onset DM2, which was therefore termed as late-onset DM2. It can be seen from Figure 6 that variations in the inflammatory markers IL-37 and IL-17A were higher than those in the inflammatory marker NLR and in TSH.

The MLR models presented in Table 5 indicate variables that significantly correlated with any of the emerging inflammatory markers, including NLR and the cytokines IL-17A and IL-37. The models had in common that all of these markers were associated with decreased renal function and with each other, but the directions of these associations were different. For example, while the values of the markers NLR and IL-17A were higher in less-graded stages of renal function decline, the values of the marker IL-37, in contrast, increased as the stages of renal function decline became higher. The marker NLR was associated positively with the cytokine IL-17A but negatively with the cytokine IL-37. The cytokines IL-17A and IL-37 were positively associated with each other, but serum concentrations of cytokine IL-37 were much more dependent on the levels of cytokine IL-17A (as indicated by the IL-37-related MLR model) compared to the cytokine IL-17A, the serum values of which were less strongly dependent on the levels of the cytokine IL-37 (as indicated by the IL-17A-related MLR model).

Interesting in these models, too, was that higher BMI and some comorbidities, such as osteoarthritis, impacted the rise in NLR values but did not affect the other two markers of inflammation. Metabolic syndrome-related and insulin resistance-related dyslipidemia, indicated by increased triglycerides and decreased HDL cholesterol, were positively associated only with the marker IL-37, while on the contrary, the markers NLR and IL-17A were higher when there were no signs of metabolic syndrome. The regulation of the markers NLR and IL-37 was associated with activation of the hypothalamic–pituitary–thyroid axis (NLR-related and IL-37-related MLR models), and this association was much stronger in the case of the marker IL-37.

## 5. Discussion

Patients with DM2 usually have multiple comorbidities, which increases interindividual variability and requires more patient-centered (personalized) management and medications [50,65]. Some comorbidities are conventionally considered as CV risk factors, some as diabetic complications, and some as accompanying (unrelated) to DM2. Recent knowledge about the common pathophysiological background of these comorbidities, with inflammation being in the center of the pathophysiological networks, has indicated that a more integrated approach is needed for classifying these disorders [2,6,65].

Distinctly from acute inflammation, for which the time course is well known, the phases of chronic inflammation that are associated with aging and the development of chronic age-related diseases are poorly identified; only some isolated mechanisms have been discerned [66,67]. The mechanisms of chronic inflammation are yet thought to show dynamics of change that run in parallel with advancement in end-organ damage and decline in whole-body entropy [3,4]. Regarding patients with DM2, the broken pieces of knowledge—how DM2 usually develops in obese individuals, how adipose tissue is a source of inflammation, how inflammatory mechanisms are implicated in the development of comorbidities, and how accumulated comorbidities exhaust homeostasis reserves and promotes frailty—need to be better integrated [9,19]. Thanks to the availability of new methodologies for data analysis, we have started to integrate this knowledge [16,17,65].

In this study, we aimed to identify the patterns of inflammatory markers that are involved in certain clinical contexts, determined by age and gender of patients with DM2 and including also information on DM2 and hypertension duration and the time of onset, as the core framework for determining these patterns. By keeping in mind that the scored values of markers of inflammation stay in a balance with the net effect of disorders that can influence their values, phenotypes indicated by the clusters provided insights into the phases of inflammatory responses that are involved in strictly defined clinical frameworks, thus being reflective of the dynamics of developing DM2-associated comorbidities [3,68].

The identified clusters can be used for risk assessment and individualized management of diabetic patients and for planning future research. In particular, in longitudinal studies, these results could show which clusters lead to which outcomes by which rates of change. Knowing these relationships would be of the utmost importance for planning prevention.

The reasoning behind the clusters is illustrated by the in-depth analysis of the three-cluster model. An advantage of this model over the six-cluster model is that it is simpler and thus enables a clearer view of how patient sociodemographic and clinical characteristics are balanced with the patterns of inflammatory markers. This analysis, e.g., showed that patients in cluster Cl 2/3, who featured female gender, obesity, age around 60, and new-onset hypertension and DM2, fit well to the evidence indicating that women usually start to gain weight and obesity-associated disorders, such as hypertension and DM2, in the years that come after menopause [69]. This all-at-once accumulation of multiple metabolic and vascular factors is expected to promote end-organ damage faster than in cluster Cl 1/3, in which patients were of similar age but had different dynamics of CV risk factor accumulation, since hypertension preceded the onset of DM2 by years. Accordingly, patients in cluster Cl 2/3 showed higher serum concentrations of the inflammatory markers NLR and IL-17A than patients in cluster Cl 1/3. In this regard, and as the evidence has also indicated, these inflammatory markers are likely to reflect the magnitude of inflammation that is evoked by inflammatory cell accumulation on the surface of dysfunctional vascular endothelial cells, and thus also the patient capacity for target tissue damage [21,31,32]. The development of CVD in these patients may yet be postponed, which is associated with the fact that patients in this cluster were dominantly women. In this regard, the evidence has suggested that women in early postmenopause have vascular endothelial cells relatively preserved from inflammatory cell adhesion compared to those of age-matched men, which can protect, for a while, target organs from inflammation-mediated damage [70,71]. Because there is a lag in time between when CV risk factors start to act on vascular endothelial cells, reverting them into being dysfunctional, and when the significant CV pathology develops, patients in cluster Cl 2/3 were not featured with marked CV comorbidities but had only initially impaired renal function (eGFR < 60) [41]. The low intensity of tissue-related inflammation in patients in cluster Cl 1/3, on the contrary, could not evoke a sufficient suppressive response (as proved by low values of the suppressive cytokine IL-37), and systemic compensatory anti-inflammatory mechanisms had to take place (as proved by higher TSH) [72].

Unlike clusters Cl 1/3 and Cl 2/3, which referred to patients with recent-onset DM2, cluster Cl 3/3 referred to patients of older age (around 70), mostly males, in whom hypertension and DM2 had been present for a long time, since their middle age. Epidemiological data have indicated that hypertension and DM2 start earlier in life in men than in women [73]. In line with their older age and long-lasting main CV risk factors, as the evidence has also suggested, patients in cluster Cl 3/3 had multiple CV comorbidities, including atherosclerotic CVD (CAD and PAD) and fairly low renal function (level 3) [13,26,74]. Specifically, it is known that men are more predisposed to atherosclerotic diseases than women [26]. Taking this all into account, it is not surprising that frailty was featured only in patients in cluster Cl 3/3. Frailty is known to develop in older individuals with multiple comorbidities, in particular CVD, CKD, and DM2 [75,76,77]. The pattern of laboratory tests fit to this clinical context, as seen by lower Hb and HDL cholesterol and higher triglycerides than in the other two clusters, which could have been the effect of lower renal function and frailty [13,38,78]. The frailty in patients in this cluster could also explain the somewhat distorted body shape, as shown by the modest “BMI” values combined with disproportionally lower “mac” values, indicating more muscle than weight loss. This can be the case when frailty develops in previously obese individuals [19,79].

As suggested by positive correlations between the marker NLR and the cytokine IL-17A and CRP, in the MLR model in which NLR was an outcome measure, during the progression phase of end-organ damage, there was a positive inflammation regulation loop. In this regard, the cytokine IL-17A is known to act as a propagator of tissue cell infiltration associated with end-organ damage [30,31,32].

In advanced phases of the end-organ damage progression, however, and when serum concentrations of the cytokine IL-17A increase to some extent, it is time for the suppressive cytokine IL-37 to take up a role. This scenario was suggested by higher values of the cytokine IL-37 in cluster Cl 2/3 than in cluster Cl 1/3 and by this cytokine being markedly increased in cluster Cl 3/3, wherein the cytokine IL-17A remained low. The positive correlations between the cytokines IL-17A and IL-37, as demonstrated in the MLR models in which either the cytokine IL-17A or the cytokine IL-37 were outcome measures, further supported this argument. In particular, in the MLR model in which the cytokine IL-37 was an outcome measure, this correlation was very strong, indicating strong dependence of the cytokine IL-37 on even small rises in serum concentrations of the cytokine IL-17A, which goes in line with the suppressor-regulatory role of the cytokine IL-37.

Further in this context, the strong positive correlation of the cytokine IL-37 with higher levels of renal function decline in the IL-37-related MLR model could be a sign of the body’s efforts to suppress the overwhelming inflammation in conditions characterized by exhaustion of homeostatic mechanisms and could be associated with frailty. In this way, our results supported the evidence indicating the cytokine IL-37 as having broad anti-inflammatory and modulatory effects [33]. One of the modes of action of this cytokine may be by decreasing the level of insulin resistance [34]. Chronic renal failure is also known to be an insulin-resistant state associated with metabolic syndrome-related disorders [80]. This in our results was supported by the lower HDL cholesterol and higher triglycerides in cluster Cl 3/3 than in the other two clusters and by the negative correlation of the cytokine IL-37 with HDL cholesterol in the IL-37-related MLR model. An intriguing hypothesis that arose from the analysis of the clusters in the three-cluster model is that activation of the hypothalamic–pituitary–thyroid axis might represent a systemic defense mechanism triggered during the progression phase of end-organ damage in order to revert the slowed-down energy metabolism and to retard the dissemination of inflammatory cells to the tissues [72,81]. This notion was supported by the higher TSH values (at the upper border of the reference range) in patients in cluster Cl 1/3 than in the other two clusters and by the negative correlation between TSH and the marker NLR in the NLR-related MLR model. When tissue-related defense mechanisms are going to become exhausted because of overwhelming tissue infiltration by inflammatory cells and fibrotic processes prevailing over inflammation processes, as in conditions associated with multiple CV comorbidities and frailty, there could be also a breakdown of the systems regulating these mechanisms. In our results, this was likely indicated by low TSH values in patients in cluster Cl 3/3 and by the negative correlations of the cytokine IL-37 with both TSH and CRP in the MLR model in which IL-37 was an outcome measure. The cytokine IL-37 was found to rise in conditions associated with fibrosis [33]. The acute-phase reactant CRP is known for its multiple inflammation-regulating roles [82].

The diversification of the cluster subtypes when the k-value increased from three to six revealed that in older individuals with DM2, mainly women contributed to the expansion of the phenotypes. A similar conclusion was drawn by some other authors on a large scale [65]. For this reason, in future studies with similar design, men and women should be considered separately. The same was indicated by the epidemiological data. In this regard, it was shown that the gender-dependent differences in the life courses of hypertension, metabolic syndrome, and DM2 influenced differences in the dynamics of CV risk factor accumulation and CVD development [69,73]. As identified in the study of Corrao et al., there are differences in the structure of comorbidities between older men and women [83]. Furthermore, in this study, sex dimorphism may have influenced the distribution of comorbidities in the clusters. Specifically, in women, as we emphasized in our previous studies, the transition from pre- to postmenopause and the early postmenopausal period are critical periods in women’s lives when CV risk factors are significantly modified, which specifically in women can add to the diversity of DM2-related phenotypes [44,68]. Some other studies have also provided important insights into the reasons for the heterogeneity of older patients with DM2, indicating factors such as the current patient age and DM2 duration and age of onset [41].

Known factors that could influence variations in inflammatory markers in patients with DM2, as evidence has suggested, include both age-dependent changes in the immune system and gender-dependent differences in inflammatory response [22,26]. Of not less importance, there could be an interdependence between the course of aging, either successful or unsuccessful, and the level of inflammation, which in great part is influenced by lifestyles and environmental factors [1,4,22]. Specifically, in women, the decline in the sex hormones that occurs during the menopausal transition may additionally influence immune and inflammatory responses [31]. Another question that deserves attention when evaluating inflammatory markers in older patients with DM2 is whether or not frailty has developed. This is important because this condition can alter, in an unexpected way, anthropometric measures and laboratory tests, including the values of inflammatory markers [79,84].

In this study, all these pieces of evidence were lumped together and helped explain variations in the selected set of inflammatory markers. We could see how different combinations of factors, such gender, age, DM2 duration and age of onset, and the effect of some treatments, were associated with some CV risk factors, body shape characteristics, and CV and non-CV comorbidities, as well as how inflammatory markers were justified among these multifaceted patterns. Although it is not possible to fully understand all associations between variations in inflammatory markers as expressed by the six-cluster model and the diversity of the identified phenotypes, we could draw some general conclusions from these obtained results. The longitudinal follow-up of patients in the clusters in associations to specific outcomes could give more detailed insights in this regard.

It was evident from the results obtained by the six-cluster model that variations in inflammatory markers that are already established in clinical practice, such as CRP and NLR, were lower than variations in the cytokines IL-17A and IL-37, which justifies the usefulness of these cytokines for routine use. The cytokine IL-37 was shown to be especially more distinctive than other explored markers. However, the fact that this cytokine increased only in some specific situations limits its use as a single deciphering marker. Furthermore, the marker NLR, if used alone, as we discussed in our previous review paper, can hardly differentiate between the subgroups of women who are in the age around menopause or in early postmenopausal period (age range 50–65 years), in spite of the fact that they may differ from each other in CV and metabolic profiles. Furthermore, this marker, when used alone, cannot differentiate between women of this age and age-matched men [31]. The results of this study indicated the same. Specifically, although the NLR scores were similar, e.g., in clusters Cl 1/6 and Cl 4/6, the characteristics of patients in these clusters were nonetheless different. When, however, the cytokine IL-17A was also considered, the higher values of this cytokine in cluster Cl 1/6 than in cluster Cl 4/6 likely indicated the higher potential of patients in cluster Cl1/6 than those in Cl4/6 to develop tissue-related inflammation progression. Taken together, the results of this study indicated that it is better to use a set of inflammatory markers, than any single one, to discern among the phenotypes of patients with DM2.

We gained further insights into how the set of inflammatory markers indicated different phases of chronic inflammation associated with DM2-related end-organ damage when analyzing clusters Cl l/6, Cl 5/6, and Cl 6/6 of the six-cluster model. These clusters had in common a prevailing female gender and a lack of CV comorbidities. They differed from each other, however, in patterns of inflammatory markers, reflecting differences in the potential of individuals in these clusters to develop end-organ damage and CV complications. In this regard, women in cluster Cl 1/6 may have been in a less favorable position than their counterparts in cluster Cl 5/6 based on the former’s higher values of the markers NLR and IL-17A, and in parallel to this, their lower values of the cytokine IL-37. This pattern of inflammatory markers in patients in cluster Cl 1/6 may be reflective of the long-term duration of both main CV risk factors, DM2 and hypertension, and/or of DM2 onset earlier in life. In this regard, other studies have shown that the potential for complications to develop in patients with DM2 is higher when the age of DM2 onset is lower [41].

In a better position than these two patient subgroups, with respect to CV prognosis, may have been women in cluster Cl 6/6. This was supported by even lower values of the markers NLR and IL-17A, albeit higher values of the cytokine IL-37, in women in this cluster than in those in the other two. This better inflammatory profile was in line with a more favorable CV risk profile of women in cluster Cl 6/6, as indicated by the lack of obesity and hypertension.

If, however, we bring together evidence indicating that the serum concentrations of the cytokine IL-37 rise in parallel to the level of insulin resistance and our results obtained by the six-cluster model, we can gain some further insights into associations between the phases of chronic inflammation and the progression rates of end-organ damage in patients with DM2 [34]. In this regard, we can see that higher values of the cytokine IL-37 appeared not only in patients in cluster Cl 4/6 but in those in clusters Cl 2/6 and Cl 6/6. Taking into account the common features of these two clusters, higher values of the cytokine IL-37 can be expected in individuals with recently acquired DM2 and who are younger than 65, which may be a basis for their high potential for developing complications, than when DM2 occurs later in life, after the age of 65 (as in cluster Cl 5/6) [41]. Seeing it this way, it may be that women in cluster Cl 6/6 had an equal or even worse CV risk than women in cluster Cl 5/6, in spite of the fact that both groups lacked CV comorbidities. Only longitudinal studies can give a clear answer to this dilemma.

As the last example for discussion, we use the two oldest patient groups in the six-cluster model, clusters Cl 3/6 and Cl 4/6. The common features of these clusters were low renal function and multiple CV and other comorbidities. However, these clusters markedly differed from each other in patterns of inflammatory markers. The values of these markers were lower in patients in cluster Cl 3/6; in fact, these values were at the lowest levels in this cluster out of all of the identified clusters. The characteristics of patients in this cluster that could explain such a pattern of inflammatory markers could be late-onset DM2 (at the age of 70) and female gender. In this regard, the evidence has suggested that the onset of DM2 in advanced age, after the age of 70 or at age 80, bears low risk of preterm mortality [41]. As we shown on the example of the marker NLR, the capacity for inflammatory cell dissemination throughout the tissues may in postmenopausal women be low [31]. The low values of inflammatory markers were not merely a sign of homeostatic mechanism exhaustion; the proof of this may be the lack of frailty in patients in cluster Cl 3/6.

In contrast to cluster Cl 3/6, the prominent feature of patients in cluster Cl 4/6 was frailty. In line with this feature, there was a much higher score value of the suppressive inflammatory marker IL-37 in patients in this cluster, indicating advanced stages of end-organ damage and vascular complication development associated with fibrotic processes prevailing over inflammatory processes [31,33,85]. This, however, may not be a bad prognostic sign, taking into account the evidence that an increase in IL-37 occurs in conditions with increased insulin resistance, which in very old males, in spite of the presence of overt atherosclerotic disease, can be a sign of longevity [5,26,34]. In this way, our results supported the view of aging as a process of remodeling rather than of straightforward decline in homeostatic reserves [86].

From the same perspective we can see inconsistent results between the three-cluster and six-cluster models relating to the values of the hormone TSH. The values of this regulatory hormone took the opposite magnitudes in two complementary clusters, Cl 1/3 in the three-cluster model and Cl 4/6 in the six-cluster model. Based on the existing knowledge and our results, and including the strong negative correlation between these two markers in the IL-37-related MLR model, there may be a strict feedback loop between the cytokine IL-37 and the hormone TSH [33,72]. In this context, the hormone TSH could be another key deciphering factor for discerning among DM2-related phenotypes, which may have also prognostic and therapeutic relevance.

A separation of older patients with DM2 into discrete phenotypic subgroups, as done in this study, may be a way of deconstructing these patients’ complexity and can be used as a model for guiding personalized treatment. This research approach could be especially beneficial for family physicians, who are in the position of dealing with the complexity of these patients on a daily basis.

For example, very old patients with CV comorbidities and frailty, as in cluster Cl 4/6, in spite of increased insulin resistance, should not be treated with insulin-sensitizer drugs, or with many medications at all. Rather, they should be provided with general care and nutritional support. For women with high values of the inflammatory markers NLR and IL-17A combined with low IL-37 values, as in cluster Cl 1/6, a solution may be an anti-inflammatory treatment or an IL-37-based therapy. For women with modest CV risk and inflammatory profiles, such as those in cluster Cl 6/6, on the contrary, only nonpharmacological and lifestyle-modifying measures should be enough. Finally, for women in cluster Cl 3/6, who might have a low potential for the advancement of CV comorbidities and who suffer from affective disorders (e.g., anxiety/depression), recommendations for physical activity and cognitive behavioral therapy may be an optimal care option. These theoretically proposed treatment protocols should, however, first be assessed by prospective evaluations.

## 6. Conclusions

DM2 is a leading public health concern associated with multiple comorbidities, of which CVD has the most important prognostic role. Initiatives are emerging for a more personalized approach to managing patients with DM2. Chronic inflammation is considered to be a major force driving the development of DM2 and its associated comorbidities. Currently, there is no consensus as to which markers of inflammation best represent which phases of chronic inflammatory response. In this study, we aimed to show that by using a set of inflammatory markers clustered with variables indicating the sociodemographic and clinical contexts of diabetic patients, it is possible to identify different, reasonable phenotypes that may exist among diabetic patients. By analyzing the intra- and intercluster patterns of inflammatory markers, it was possible to recognize clinical contexts in which different mechanisms of chronic inflammation were involved. The approach presented in this paper might offer a reliable new strategy for personalized risk stratification and management of diabetic patients. More generally, if implemented in everyday practice of PC physicians, this research approach could promote the development of new, integrated strategies for managing older individuals with multiple chronic conditions.

## Figures and Tables

**Figure 1 healthcare-09-01687-f001:**
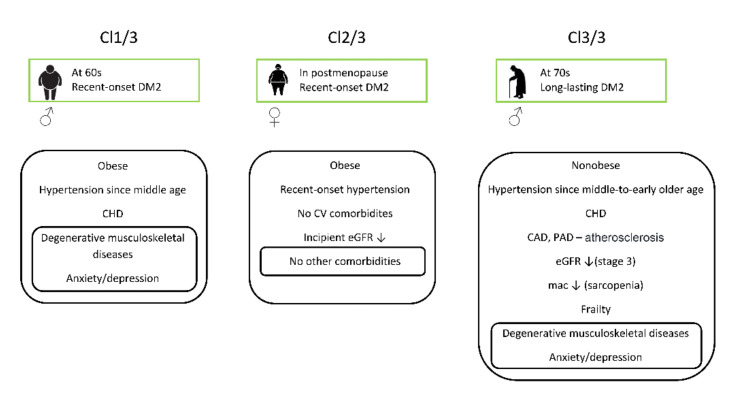
Sociodemographic and clinical contexts of patients in the clusters identified by the 3-cluster model. Cl—cluster; DM2—diabetes mellitus type 2; mac—midarm circumference; CV—cardiovascular; CHD—chronic heart disease; CAD—coronary artery disease; PAD—peripheral artery disease; eGFR—estimated glomerular filtration rate.

**Figure 2 healthcare-09-01687-f002:**
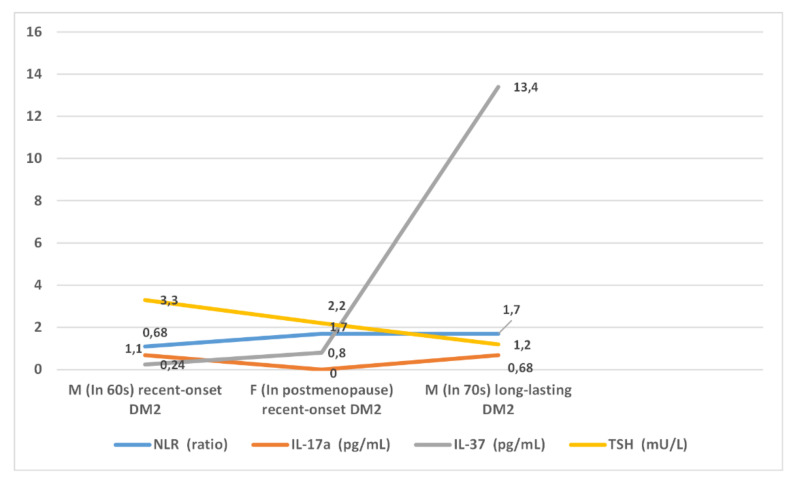
A graphical presentation of the inflammatory markers NLR, IL-17A, and IL-37, and the hormone TSH across the clusters in the three-cluster model. M—males; F—females; DM2—diabetes mellitus type 2; NLR—neutrophil-to-lymphocyte ratio; Il-17A—cytokine IL-17A; IL-37—cytokine IL-37; TSH—thyroid-stimulating hormone.

**Figure 3 healthcare-09-01687-f003:**
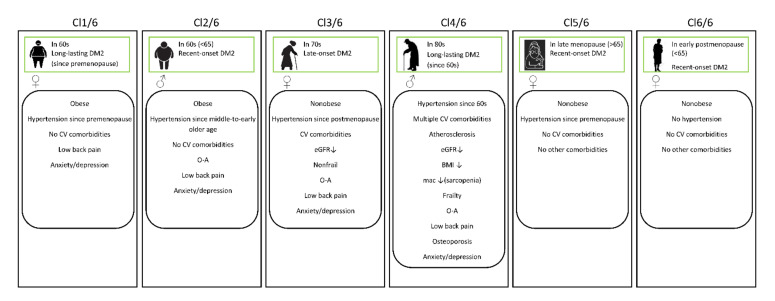
Sociodemographic and clinical contexts of patients in the clusters identified by the 6-cluster model. Cl—cluster; DM2—diabetes mellitus type 2; BMI—body mass index; mac—midarm circumference; CV—cardiovascular; O-A—osteoarthritis; eGFR—estimated glomerular filtration rate.

**Figure 4 healthcare-09-01687-f004:**
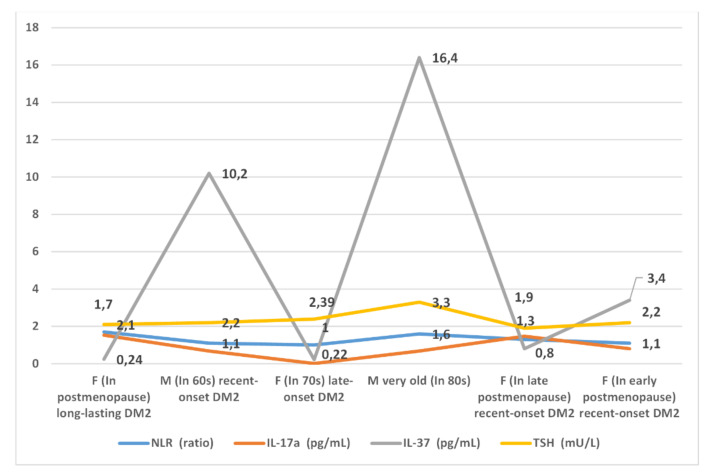
The graphical presentation of inflammatory markers NLR, IL-17A, and IL-37, and the hormone TSH, across the clusters, in the 6-cluster model. M—males; F—females; DM2—diabetes mellitus type 2.

**Figure 5 healthcare-09-01687-f005:**
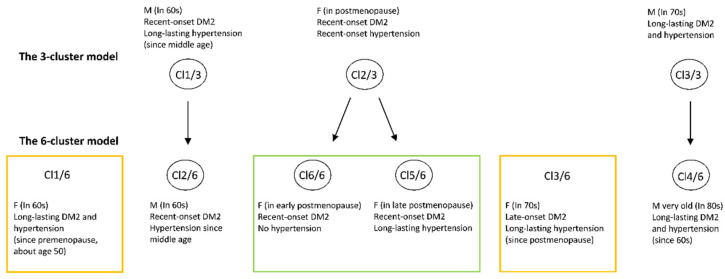
Diversification of the phenotypes when the number of clusters increased from 3 to 6. Clusters framed in green color, in the 6-cluster model, are proposed to be derived from the “parent” clusters in the 3-cluster model; Clusters Cl 1/6 and Cl 3/6, framed in yellow color, were newly identified subtypes. Cl—cluster; M—males; F—females; DM2—diabetes type 2.

**Figure 6 healthcare-09-01687-f006:**
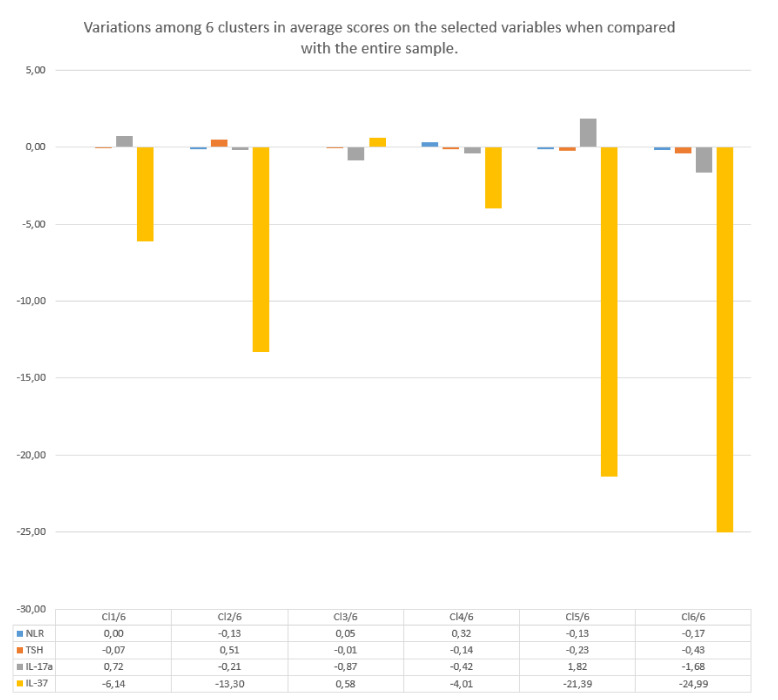
Variations in inflammatory markers NLR, IL-17A, and IL-37 and in the hormone TSH among the clusters in the 6-cluster model. Cl—cluster; NLR—neutrophil-to-lymphocyte ratio; Il-17A—cytokine IL-17A; IL-37—cytokine IL-37; TSH—thyroid-stimulating hormone. The average scores of the selected variables were transformed in a way to show the justified units, which could be be above or below the average of the sample.

**Table 1 healthcare-09-01687-t001:** The variable description for 30 variables included in the study.

No	Variables and Their Abbreviations	Variable Description	
1	Gender	Categorical (0,1) Category frequency Males = 98 Females = 76	
2	Age (years)	Numerical Mean: 67.03 Max = 88, Min = 50	
3	Number of comorbidities (<3, ≥3)	Categorical (0,1) Category frequency 0 = 8 1 = 166	
4	Body mass index (BMI) (kg/m^2^)	Numerical Mean: 30.53 Max = 51.37, Min = 19.4	
5	Midarm circumference (mac) (cm)	Numerical Mean: 29.974 Max = 39, Min = 21	
6	Marker of inflammation-related anemia Hemoglobin (Hb) (g/L)	Numerical Mean: 142.27 Max = 166, Min = 93	
7	Estimated glomerular filtration rate (eGFR) (mL/min/1.73 m^2^)	Numerical Mean: 80.723 Max = 163, Min = 29	
8	eGFR levels	Nominal 1,2,3,4	
		Category	Frequency
		1	55
		2	64
		3	43
		4	12
9	Triglycerides (mmol/L)	Numerical Mean: 1.981 Max = 8.7, Min = 0.53	
10	HDL cholesterol (mmol/L)	Numerical Mean: 2.209 Max = 141, Min = 2.49	
11	Thyroid-stimulating hormone (TSH) (mU/L)	Numerical Mean: 2.905 Max = 9.45, Min = 0.15	
12	Traditional marker of inflammation C-reactive protein (CRP) (mg/L)	Numerical Mean: 2.877 Max = 27.2, Min = 0.2	
13	Frailty index	Categorical 0,1,2	
		Category	Frequency
		0	101
		1	42
		2	31
14	Diabetes mellitus type 2 (DM2) duration (years)	Numerical Mean: 8.809 Max = 30, Min = 1	
15	Hypertension	Categorical (0 = No, 1 = Yes) Category frequency 0 = 19 1 = 155	
16	Hypertension duration (years)	Numerical Mean: 10.48 Max = 25, Min = 0	
17	Diagnosis of chronic heart disease (CHD)	Categorical (0 = No, 1 = Yes) Category frequency 0 = 92 1 = 82	
18	Diagnosis of coronary artery disease (CAD)	Categorical (0 = No, 1 = Yes) Category frequency 0 = 114 1 = 60	
19	Diagnosis of periphery artery disease (PAD)	Categorical (0 = No, 1 = Yes) Category frequency 0 = 137 1 = 37	
20	Diagnosis of osteoporosis	Categorical (0 = No, 1 = Yes) Category frequency 0 = 135 1 = 39	
21	Diagnosis of severe osteoarthritis	Categorical (0 = No, 1 = Yes) Category frequency 0 = 89 1 = 85	
22	Diagnosis of low back pain	Categorical (0 = No, 1 = Yes) Category frequency 0 = 115 1 = 59	
23	Diagnosis of anxiety/depression	Categorical (0 = No, 1 = Yes) Category frequency 0 = 105 1 = 69	
24	New treatment option Dipeptidyl peptidase-4 inhibitor (DPP4)	Categorical (0 = No, 1 = Yes) Category frequency 0 = 143 1 = 31	
25	New treatment option Sodium glucose cotransporter-2 inhibitors (SGLT2)	Categorical (0 = No, 1 = Yes) Category frequency 0 = 167 1 = 7	
26	New treatment option Glucagon-likepeptide-1 receptor agonists (GLP1r)	Categorical (0 = No, 1 = Yes) Category frequency 0 = 158 1 = 16	
27	Therapy with insulin	Categorical (0 = No, 1 = Yes) Category frequency 0 = 134 1 = 40	
28	Emerging marker of inflammation Neutrophil-to-lymphocyte ratio (NLR)	Numerical Mean: 1.718 Max = 5.4, Min = 0.6	
29	Emerging marker of inflammation Il-17A (pg/mL)	Numerical Mean: 3.291 Max = 75.52 Min = 0.01	
30	Emerging marker of inflammation Il-37 (pg/mL)	Numerical Mean: 42.679 Max = 1788.4 Min = 0.14	

**Table 2 healthcare-09-01687-t002:** Cluster analysis evaluation.

K Value	Number of Iterations	Within-Cluster Sum of Squared Errors
3	8	2712
4	7	2689
5	8	2593
6	7	2592

**Table 3 healthcare-09-01687-t003:** Clusters obtained by k-means algorithm with k = 3. The 3-cluster model.

Variable	Cl 1/3	Cl 2/3	Cl 3/3
Gender (M,F) (0,1)	1	0	1
Age (years)	61	63	72
Number of comorbidities (<3, ≥3) (0,1)	1	1	1
BMI (kg/m^2^)	37	32	27
mac (cm)	29	30	28
Hb (g/L)	148	142	123
eGFR (mL/min/1.73 m^2^)	82	59	29
eGFR levels (1–4)	2	1	3
Triglycerides (mmol/L)	1.5	1.6	2.1
HDL cholesterol (mmol/L)	1.4	1.3	1.2
TSH (mU/L)	3.3	2.2	1.2
CRP (mg/L)	1.2	1.1	0.8
Frailty index (0,1,2)	0	0	2
DM2 duration (years)	1	2	10
Hypertension (0 = No, 1 = Yes)	1	1	1
Hypertension duration (years)	10	0	15
CHD (0 = No, 1 = Yes)	1	0	1
CAD (0 = No, 1 = Yes)	0	0	1
PAD (0 = No, 1 = Yes)	0	0	1
Osteoporosis (0 = No, 1 = Yes)	0	0	0
Severe osteoarthritis (0 = No, 1 = Yes)	1	0	1
Low back pain (0 = No, 1 = Yes)	1	0	1
Anxiety/depression (0 = No, 1 = Yes)	1	0	1
DPP4 therapy (0 = No, 1 = Yes)	0	0	0
SGLT2 therapy (0 = No, 1 = Yes)	0	0	0
GLP1r therapy (0 = No, 1 = Yes)	0	0	0
Insulin therapy (0 = No, 1 = Yes)	0	0	0
NLR	1.1	1.7	1.7
Il-17A (pg/mL)	0.68	1.42	0.68
Il-37 (pg/mL)	0.24	0.8	13.4

Cl—Cluster; BMI—body mass index; mac—midarm circumference; Hb—hemoglobin; NLR—neutrophil-to-lymphocyte ratio; eGFR—estimated glomerular filtration rate; TSH—thyroid-stimulating hormone; CRP—C-reactive protein; DM2—diabetes mellitus type 2; CHD—chronic heart disease; CAD—coronary artery disease; PAD—peripheral artery disease; DPP4—dipeptidyl peptidase-4 inhibitor; SGLT2—sodium-glucose cotransporter-2 inhibitors; GLP1r—glucagon-like peptide-1 receptor agonists.

**Table 4 healthcare-09-01687-t004:** Clusters obtained by k-means algorithm with k = 6. The six-cluster model.

Variable	Cl 1/6	Cl 2/6	Cl 3/6	Cl 4/6	Cl 5/6	Cl 6/6
Gender (M,F) (0,1)	0	1	0	1	0	0
Age (years)	61	64	72	80	67	61
Number of comorbidities (<3, ≥3) (0,1)	1	1	1	1	1	1
BMI (kg/m^2^)	32	31.64	28.4	24.03	26	26.78
mac (cm)	32	30	33	27	29	28
Hb (g/L)	157	142	123	134	129	162
eGFR (mL/min/1.73 m ^2^)	79	87	29	59	59	58
eGFR levels (1–4)	1	2	3	3	2	1
Triglycerides (mmol/L)	1.5	1.2	1.2	1.9	2.1	1
HDL cholesterol (mmol/L)	1.3	1.2	1.2	1.2	1.5	1.6
TSH (mU/L)	2.1	2.2	2.39	3.3	1.9	2.2
CRP (mg/L)	1.1	0.9	0.6	0.8	1.1	1.2
Frailty index (0,1,2)	0	0	0	2	0	0
DM2 duration (years)	10	2	1	12	3	2
Hypertension (0 = No, 1 = Yes)	1	1	1	1	1	0
Hypertension duration (years)	10	7	10	15	15	0
CHD (0 = No, 1 = Yes)	0	0	1	1	0	0
CAD (0 = No, 1 = Yes)	0	0	1	1	0	0
PAD (0 = No, 1 = Yes)	0	0	1	1	0	0
Osteoporosis (0 = No, 1 = Yes)	0	0	0	1	0	0
Severe osteoarthritis (0 = No, 1 = Yes)	0	1	1	1	0	0
Low back pain (0 = No, 1 = Yes)	1	1	1	1	0	0
Anxiety/depression (0 = No, 1 = Yes)	1	1	1	1	0	0
DPP4 therapy (0 = No, 1 = Yes)	0	0	0	0	0	0
SGLT2 therapy (0 = No, 1 = Yes)	0	0	0	0	0	0
GLP1r therapy (0 = No, 1 = Yes)	0	0	0	0	0	0
Insulin therapy (0 = No, 1 = Yes)	0	0	0	0	0	0
NLR	107	1.1	1	1.6	1.3	1.1
Il-17A (pg/mL)	1.53	0.68	0.01	0.68	1.47	1.42
Il-37 (pg/mL)	0.24	10.2	0.22	16.4	0.8	3.4

Cl—cluster; BMI—body mass index; mac—midarm circumference; Hb—hemoglobin; NLR—neutrophil-to-lymphocyte ratio; eGFR—estimated glomerular filtration rate; TSH—thyroid-stimulating hormone; CRP—C-reactive protein; DM2—diabetes mellitus type 2; CHD—chronic heart disease; CAD—coronary artery disease; PAD—peripheral artery disease; DPP4—dipeptidyl peptidase-4 inhibitor; SGLT2—sodium-glucose cotransporter-2 inhibitors; GLP1r—glucagon-like peptide-1 receptor agonists.

**Table 5 healthcare-09-01687-t005:** Multivariate linear regression analysis for NLR, Il-17A, and Il-37 variables as dependent variables.

NLR = −0.2602 × Gender + 0.0156 ∗ BMI + −0.0077 × eGFR in mL/min/1.73m^2^ + −0.058 × eGFR levels + −0.0813 ∗ Triglycerides + −0.081 × TSH + 0.0182 × CRP + 0.3491 × Severe osteoarthritis + 0.0149 × Il-17A + −0.0007 × Il-37 + 2.2697
Il-17A = 0.0584 × age in years + −1.1963 × eGFR levels + 0.0459x − xHDL + 0.3422 × CRP + 0.0385 × Il-37 + −0.8856
Il-37 = 17.377 × eGFR levels + −0.883 × HDL + −6.4164 × TSH + −4.9867 × CRP + 17.3077 × Il-17A + −15.2037

BMI—body mass index; NLR—neutrophil-to-lymphocyte ratio; eGFR—estimated glomerular filtration rate; TSH—thyroid-stimulating hormone; CRP—C-reactive protein; HDL—HDL cholesterol; IL-17A and IL-37—cytokines.

## Data Availability

Not applicable.

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
