# Peer review of "Clustering Inflammatory Markers with Sociodemographic and Clinical Characteristics of Patients with Diabetes Type 2 Can Support Family Physicians’ Clinical Reasoning by Reducing Patients’ Complexity"

_healthcare, 2021, doi:10.3390/healthcare9121687_

Round 1

Reviewer 1 Report

Reviever Report

The manuscript presents an attractive viewpoint toward personalized medicine by highlighting the importance of recognizing patient profiles, their characteristics, and finding targeted treatments. I greatly appreciate the authors' work. However, before further consideration of this manuscript for possible publication, the authors should consider the following points:

  1. In the Discussion, it is difficult to follow the cluster numbers; it is confusing whether they refer to the 3-cluster model or to the 6-cluster model. It would be better to assign the numbers to the model, e.g., "Cluster 2/3" and "Cluster 1/6", ...

  1. The reference to Fig. 1 (in line 450) is a bit late. It would be better to move this part of the discussion to the Results part; it would be a great help for the reader if the results were presented with Fig. 1, which is actually also the main result.

  1. There is only one figure in the whole article, Fig. 1, and it is rather uninformative. It would be good to add the main characteristics of the clusters in Fig. 1. Please add the trends of the three main parameters: NLR, IL -17A, and IL -37 to the clusters, and write either that they are lower/decreasing or high/increasing. This is the most important result and should also be shown graphically. The authors could also reconsider the graphical representation and think about some kind of graphical abstract/summary of the results.

  1. The discussion should indicate whether these results could also be useful for longitudinal studies.

  1. The conclusions should make clear how clinicians can use these results in clinical practice. The abstract and introduction, and even the title, promise much in term of results that can be used by primary care physicians.

  1. In lines 327-329, the authors write, "The metabolic syndrome-related and insulin resistance-related dyslipidemia, indicated by elevated triglycerides and decreased HDL cholesterol, is an influential variable for all the emerging markers of inflammation." What results of the authors' study support this finding? Please specify very clearly based on the results presented in the tables.

  1. Methods: In the Discussion, please add more detail on what the results look like when the number of clusters, k, is further increased; is it a consistent separation of subtypes; do all the subclusters stay in the "parent cluster" when k is increased, or do the subclusters for higher k spread out to the other clusters. Perhaps an additional figure, similar to Fig. 1, would be helpful for higher k-s.

Author Response

In attachment, we are sending a reply to report round 1 

Reviewer 2 Report

This study provides a unique approach to the analysis of type 2 diabetes mellitus, which has serious comorbidities due to a variety of individual conditions.

This is a very significant paper for the future precision medicine with type 2 diabetes.

Major;

Although many cytokines and inflammasomes have been reported to be involved in the pathogenesis of type 2 diabetes and chronic inflammation, there seems to be a lack of explanation as to why the authors focused on only IL-17A and IL-37 in this study.

Author Response

In attachment, we are sending reply to report round 1

Round 2

Reviewer 1 Report

The manuscript has been considerably improved. I recommend acceptance of the manuscript with one small comment. I would recommend that the authors use the designation for clusters consistently throughout the text and in all figures, e.g., in Fig. 5 and in the text below it is a mixture of old and new designations for clusters.

Author Response

Cover Letter for the second round of the review process

Dear Editor and the Reviewer No. 1,

We kindly ask you to accept our apologies for not fulfilling your requirements fully in the first round of the revision process.  We had to ask the computer designer to make corrections on the figures, and she was busy at that time with some other tasks.

Anyway, we have made the required changes this time. We used the identical coding for clusters in all tables, figures, their legends, and throughout the text, including the Results section, and the Discussion section.

We named a particular cluster as, e.g., cluster Cl 1/3, or cluster Cl 1/6, with the first number denoting the cluster order, and the second number denoting the cluster-based model (either it is the 3-cluster model, as in the phrase Cl 1/3, or the 6-cluster model, as in the phrase Cl 1/6).

We made the mentioned corrections in Table 3, Table 4, Fig. 1, 3, 5 and 6.

We hope that we have now met all of your expectations regarding the correctness of our manuscript and that there are no further barriers for its acceptance for publication.

On belhaf of the author`s team,

Ljiljana Trtica Majnarić, Ph.D., assoc. prof.
